# Research on Top Archer’s EEG Microstates and Source Analysis in Different States

**DOI:** 10.3390/brainsci12081017

**Published:** 2022-07-31

**Authors:** Feng Gu, Anmin Gong, Yi Qu, Hui Xiao, Jin Wu, Wenya Nan, Changhao Jiang, Yunfa Fu

**Affiliations:** 1School of Information Engineering, Engineering University of People’s Armed Police, Xi’an 710086, China; gufengpap123@163.com (F.G.); quuuyishandong@163.com (Y.Q.); 2School of Military Basic Education, Engineering University of People’s Armed Police, Xi’an 710086, China; xiaohuipap@163.com; 3Department of Physical Education, Beijing City University, Beijing 100094, China; wujinbsu@163.com; 4Department of Psychology, College of Education, Shanghai Normal University, Shanghai 201418, China; wynan1985@126.com; 5Key Laboratory of Sports Performance Evaluation and Technical Analysis, Capital Institute of Physical Education, Beijing 100088, China; jch_1226@126.com; 6School of Automation and Information Engineering, Kunming University of Science and Technology, Kunming 650032, China; fyf@ynu.edu.cn

**Keywords:** EEG microstate, archery, elite archer, sLORETA, alpha rhythm, resting state networks

## Abstract

The electroencephalograph (EEG) microstate is a method used to describe the characteristics of the EEG signal through the brain scalp electrode potential’s spatial distribution; as such, it reflects the changes in the brain’s functional state. The EEGs of 13 elite archers from China’s national archery team and 13 expert archers from China’s provincial archery team were recorded under the alpha rhythm during the resting state (with closed eyes) and during archery aiming. By analyzing the differences between the EEG microstate parameters and the correlation between these parameters with archery performance, as well as by combining our findings through standardized low-resolution brain electromagnetic tomography source analysis (sLORETA), we explored the changes in the neural activity of professional archers of different levels, under different states. The results of the resting state study demonstrated that the duration, occurrence, and coverage in microstate D of elite archers were significantly higher than those of expert archers and that their other microstates had the greatest probability of transferring to microstate D. During the archery aiming state, the average transition probability of the other microstates transferring to microstate in the left temporal region was the highest observed in the two groups of archers. Moreover, there was a significant negative correlation between the duration and coverage of microstates in the frontal region of elite archers and their archery performance. Our findings indicate that elite archers are more active in the dorsal attention system and demonstrate a higher neural efficiency during the resting state. When aiming, professional archers experience an activation of brain regions associated with archery by suppressing brain regions unrelated to archery tasks. These findings provide a novel theoretical basis for the study of EEG microstate dynamics in archery and related cognitive motor tasks, particularly from the perspective of the subject’s mental state.

## 1. Introduction

Archery is a fine sport requiring high precision and accuracy, and its performance is defined by the ability to accurately shoot an arrow at a given target [1]. Key determinants of archery performance include not only motor skills (such as strength, endurance, balance, intermuscular coordination, rhythm, and accuracy) [1,2] but also the psychological factors of concentration, relaxation, and different types of attention accompanied by visual focusing [3]. A large number of researchers in the field of archery believe that archery is mainly psychologically driven [4,5,6]. Brain activity is believed to be a direct reflection of psychological changes, particularly as the brain performs complex processes to recognize stimuli, select and plan responses, make decisions, and prepare and/or execute actions [7,8].

Previous scholars have conducted research on the neural activities underlying archery. Salazar et al. have recorded the heart rate and electroencephalograph (EEG) of the left and right temporal regions of 28 elite archers during a session of 16 shots. They found no deceleration of the heart rate, more significant power changes in the alpha waves of the left hemisphere, and no significant power changes in the right hemisphere during targeting [9]. Landers et al. (1991) have performed three modes of modes of correct, incorrect, and no feedback control biofeedback training on 24 pre-elite archers in an attempt to investigate whether the EEG biofeedback training could effectively improve archery performance by collecting EEG data from the left and the right temporal regions. The results showed that the correct feedback group significantly improved shooting performance, while the wrong feedback group did the opposite, with no significant difference in the control group’s performance. The conclusion of this study provides some support for using the relationship between EEG and shooting performance as an effective biofeedback means to affect the performance of pre-elite archers [10]. Lee (2009) has used single-channel EEG to test elite, mid-level, and novice archers’ ability to focus and to relax control during the shooting process. The study found that elite archers have shown increases in both attention and relaxation, as well as a higher level of attention at the arrow release point. Interestingly, in the same study, mid-level archers have exhibited more attention but less relaxation [11]. A study comparing the arousal levels of archers before and after a neurofeedback intervention has identified some statistically significant changes in the sensorimotor rhythm (SMR)/theta ratio of archers who had received neurofeedback training after the competition [12]. Vrbik et al. (2015) have investigated the possible differences between recurve and compound shooters in terms of named values that were investigated according to arrow scores by measuring EEG data and heart rates during shooting performed by eight experienced archers. Their study concluded that compound shooters were able to achieve higher arrow score values, and they also had higher heart rates and attention values throughout the shooting [3].

In conclusion, the use of EEG technology has become the mainstream approach to analyze the brain neural mechanisms associated with archery, and through this approach, several breakthroughs have been achieved. However, most of the aforementioned studies are based on the time and the frequency domain characteristics of the EEG and the event-related potential (ERP) signals [13,14]. As such, these analyses often need to define the region of interest (ROI) of a limited number of electrodes [15] and fail to utilize the rich spatial information inherent in the EEG/ERP signals [16]. An EEG microstate analysis based on scalp topographic map clustering can effectively remedy this defect [17]. EEG microstates are an ideal method for the study of large-scale brain network dynamics in individuals undertaking cognitive activities, and this study can be undertaken with a precision defined at the millisecond (ms) scale. Previous studies have demonstrated that although scalp topographic maps of an individual spontaneous EEG activity appear to be disorganized, scalp topographic structures remain relatively stable over short periods of time [18], and these brief periods of stability typically last 80–120 ms [19,20]. Thus, the microstate can be defined as a brief period of time during which the overall brain electrical activity remains semi-stable. Moreover, it is determined by the topographical characteristics of the potential as recorded by a multichannel electrode array, and the characteristics of each microstate are the unique topographical characteristics of the same potential across the entire channel array [17]. Researchers have explored the association between the resting EEG microstates and functional magnetic resonance imaging (fMRI) by recording both EEG and fMRI signals [21]; they found that there are four archetypal EEG microstates during the resting states, and these could explain more than 80% of the data [22,23].

Presently, the EEG microstate analysis is mainly used in the study of schizophrenia, dementia, depression, panic disorders, and other neuropsychiatric diseases [17,22,23,24]. Considerable early results were provided for its potential clinical value by detecting neurophysiological damage in diseases and by characterizing the neurophysiological changes occurring after certain interventions [25]. Some studies have also explored the associations between changes in the brain’s behavioral states unrelated to disease and specific microstate dynamics [26]. For example, sleepiness and rapid sleep have been shown to exhibit shorter microstate durations when compared with relaxed wakefulness, whereas sleepiness has been associated with more distinct microstate brain maps [27]. Microstates in fatigue have also shown significantly greater amplitude than those in alertness [28]. As people age, the duration of the microstates becomes shorter, and their occurrence becomes more frequent [29]. Additionally, previous studies have found a relationship between the emergence of microstates and specific information processing functions in studies focusing on task-oriented brain activities [25,30,31] and have revealed a relationship between motor imagination and cognitive tasks with neural states that seems to take place through the change of microstate parameters [32,33,34].

The above studies indicate that different microstates are caused by different brain neuron activities. As a representation of the global functional state of the brain, the functional state of the brain will also change when the microstate map changes. Although significant progress has been made regarding the study of the EEG analysis by using microstates, there are only a limited number of studies focusing on the application of these technologies in the field of sports science. Archery is a kind of fine movement sport requiring cognitive control, and the neural state of the brain is constantly changing when performing the act. By applying EEG microstate technology, the archer’s EEG signal can be intuitively simplified to the time series parameters in which these microstates appear alternately. The functional interpretation of these microstate parameters and the analysis of the characteristics of brain activity related to the archery task are helpful to reveal the archer’s neural state during the rest and task.

Additionally, previous studies on archery have mostly focused on the neural mechanisms related to the archery process, and few studies have actually explored the brain electrical activity of the archers during the resting state. The neural compartments of the resting brain are also active and contain a lot of valuable information [34,35,36]. The rapid transition of each microstate in the resting state indicates a rapid switch between the various neural system activities in the brain [34]. Previous resting EEG studies that have examined cognitive motor tasks have demonstrated this rapid switch in terms of neuroplasticity and neural efficiency [37,38,39]. Neuroplasticity theory believes that long-term training will lead to permanent changes in the synaptic relationship between specific functional neurons in athletes’ brains [40]. The neural efficiency hypothesis means that compared with novices, expert athletes can achieve higher performance with less neural consumption when completing their familiar sports, that is, there is a “high efficiency” phenomenon of low input and high output [41]. Meanwhile, as archery is a competitive sport [42], professional athletes and coaches are more interested in the physiological characteristics of high-level athletes with long-term training. Particularly, expert archers often undergo a lot of training to acquire master archery skills, but it is difficult for them to break through the technical bottleneck and to further grow into elite athletes of higher levels. The most important factor restricting these expert athletes from becoming elite athletes is the psychological factor. Studies have shown that for professional shooters who have mastered their shooting skills, only 20% of their performance is determined by biomechanical factors, whereas 80% is attributed to psychological factors [43]. Thus, it is of high practical significance to be able to explore the neural activity differences among professional archers demonstrating different competitive abilities in archery tasks.

In this study, we attempted to explore neurophysiological characteristics by comparing the differences of the resting state EEG microstate dynamics between elite and expert archers who have been trained for a long time. At the same time, the microstates of the cortex of archers during the preparation stage of the shooting were analyzed to further understand the dynamic changes of the neural states corresponding to archery behavior over time, as well as the close correlation between the indicators of archery behavior. Our study has assumed that the EEG microstate parameters of professional archers of different competitive levels are significantly different between the resting state and the aiming period and that these microstate parameters are closely correlated to the archery performance. In an attempt to test this hypothesis, we recorded the EEG signals of 13 national archery team athletes (elite group) and 13 provincial archery team athletes (expert group) during the resting state and the aiming period. Subsequently, we calculated the microstate parameters (such as the duration, the occurrence, the coverage, and the transition probability) as state parameters. By testing the difference between these microstate parameters and the correlation between these parameters and archery performance, the two groups of subjects in different states have exhibited statistically significant results. According to previous studies, microstates are mainly related to the alpha band (8–12 Hz) activities [44,45,46]. This rhythm has also been shown to have an important effect on exercise performance [9,47,48]. Thus, this study also focused on alpha rhythm. Source localization is an effective means to obtain the spatial information of microstate categories in the brain [49,50]. As a mature method to solve inverse problems, standardized low resolution brain electromagnetic tomography (sLORETA) has been widely used in the research field of EEG and brain computer interface (BCI) [51,52,53,54]. In this paper, we use this method to locate the source of each microstate class and to determine the spatial location of these microstates in the cerebral cortex, to combine these data with those of the physiological function of the brain regions where the microstates originate. The significant findings identified under the examined states were subjected to further analysis.

## 2. Methods

### 2.1. Subjects

This study recruited a total of 31 archers as experimental subjects. Among them, there were 16 archers from the Chinese National Archery Team (ten males, six females; age: 23 ± 5 years); these athletes were classified as masters or international masters, and their average training age was 8.1 ± 2.0 years. There were 15 archers from the Beijing Archery Team (nine males, six females; age: 23 ± 5 years); these athletes were classified as national first- or second-class athletes, and their average training age was 4.3 ± 1.5 years. The average archery performance of archers in the Chinese National Archery Team was 8.7 ± 0.5. The average archery performance of archers in Beijing Archery Team was 6.8 ± 1.4. We used a t-test to evaluate the archery performance and training age of archers on the Chinese National Archery Team and the Beijing Archery Team, respectively. These results showed that the archery performance and training age of the Chinese National Archery Team were significantly higher than that of the Beijing Archery Team (performance: *p* = 3.8 × 10^−5^, training age: *p* = 2.2 × 10^−6^). Therefore, we set the archers of the National Archery Team and the Beijing Archery Team as the “elite group” and “expert group”, respectively. Meanwhile, we also found that there were no significant differences regarding age and sex between the two groups (age: *p* > 0.05, sex: *p* > 0.05). Table 1 summarizes the demographic, training, and archery performance data regarding these two groups of subjects.

All subjects participated in archery training at least 4 days a week and for at least 6 h a day before the experiment. All subjects were right-handed, with their left hand holding the bow and their right hand hanging the string. Among them, three participants of the elite group and two of the expert group had been subjected to vision correction. After the performed correction, all subjects had normal visual acuity. The subjects had suffered from no major head injury, had never had a craniotomy, had no record of mental disease, and had an overall good physical function. None of the subjects consumed any alcohol, coffee, tea, or other stimulant drinks within 24 h before the experiment, and none of them had consumed any neurogenic drugs that could interfere with the study’s results. The experimental site was the outdoor range of the National Archery Team, and the experiment was conducted under the guidance of professional coaches. The experiment was reviewed by the Ethics Committee of the Capital Institute of Physical Education. All subjects understood the content of the experiment and signed the consent-providing agreement prior to their participation in the experiment.

### 2.2. Signal Acquisition

The EEG was performed by using a SAGA 32 channel EEG amplifier manufactured by TMSI (Oldenzaal, The Netherlands), as the latter is portable and meets the mobility requirements of this experiment. The electrode placement was performed according to the international 10–20 system, comprising 32 electrodes. The electrode positions were Fp1, Fpz, Fp2, F7, F3, Fz, F4, F8, FC5, FC1, FC2, FC6, A1, T7, C3, Cz, C4, T8, A2, CP5, CP1, CP2, CP6, P7, P3, Pz, P4, P8, POz, O1, Oz, and O2, with a ground electrode placed at the forehead. A1 and A2 as reference electrodes were placed at the left and right mastoid processes, respectively, and the average values of the two mastoid processes were taken as reference. A detailed electrode placement scheme is provided in Figure 1, and the sampling frequency was set at 500 Hz. Before the start of the experiment, the impedance of all electrodes was adjusted to keep it below 5 kΩ, and then the resting state EEG data, as well as those of the entire shooting process, were collected.

In the experiment, the EEG signal acquisition of the same task was completed for all subjects. First, the subjects’ resting state EEG was collected, with the subjects being seated on a soft, comfortable seat and relaxed. During the resting state collection, the subjects were asked not to recall anything deliberately and to keep their eyes closed for 3 min and then open for 3 min separately. Subsequently, the EEG signals were collected during the entire archery process. The athletes used their familiar bows and arrows to aim at the standard target paper of the international archery competition, placed 70 m away. The size of the target paper was 52 cm × 52 cm, the diameter of the 10 rings was 10 cm, the edges of the 10 rings extended outward every time, and the edges of the 10 rings extended 5 cm outward for nine, eight, seven, and six rings in sequence. Each time the athlete executed a shooting, the target reporter provided feedback on the shooting result after the firing, and the archer adjusted the aiming point according to the result. The participating athletes performed the shooting at their own pace, and each performed a total of 35 shots. The archery shooting time was recorded by an automatic infrared sensing system, and the time point of the firing was marked for the evaluation of the EEG signal [51]. According to the target paper, the shooting score was recorded as 6–10 points (0 points for missing the target). The archery process for each athlete was conducted independently, and each athlete was not aware of the result of other athletes. Before the experiment, the athletes were told not to bother about the result but to focus on their archery skills.

### 2.3. Signal Preprocessing

Figure 2 summarizes the data analysis process followed by this study. The collected EEG signals were transmitted to a computer for offline processing through the MATLAB R2014a platform. To minimize the individual differences among the subjects, we applied the individual alpha frequency (IAF) method for determining the alpha frequency band of different subjects. IAF refers to the frequency band between 8 and 12 Hz [38]. In this study, the Fast Fourier Transform method was used to calculate the average power position of the highest frequency band of the occipital electrodes (O1, O2, and Oz) between 8 and 12 Hz (frequency resolution 0.5 Hz) as the IAF of the subjects in the resting state with their eyes closed. According to the IAF of each subject, an iAF−2–IAF+2 (Hz) was considered as the alpha frequency.

During the EEG signal preprocessing, the finite impulse response (FIR) filter with order 1000 was firstly used for bandpass filtering of 0.1–50 Hz for all signals. Subsequently, a FIR filter with an order of 200 was used to perform band-pass filtering on all resting state signals and signals of −5 s~+2 s during the aiming period (with firing time as zero), by considering the alpha frequency of each subject on the basis of IAF as the frequency band range. Following this step, a common average reference was applied to the filtered data to eliminate the error caused by the change of the reference electrode. After that correction and by using the EEGLAB toolbox to visually assess whether the EEG was affected by the artifacts, an independent component analysis (ICA) was used to remove electrooculogram artifacts from each subject. Then, the data of the resting state and of the aiming state were segmented in terms of their time domain. The EEG data of the resting state with closed eyes were divided into 2 s fragments as a period to obtain data during the resting state of 90 trials for each subject. As for the EEG data during the archery aiming, previous scholars had claimed that the brain activity in the last few seconds before the arrow firing had the greatest influence on the archery performance [51,55]. Considering the rapid changes in the neural activity of the archers during the aiming in our experiment, as well as the fact that they could complete a shooting every 5 s on average, we analyzed the EEG data from 3 s before the shooting to the moment of firing and have considered this fragment as one trial. Through the signal preprocessing, the data of three elite archers (elite5, 8, 13) and one expert archer (expert8) were removed for the study because they demonstrated an excessive number of artifacts. The data of another expert archer whose low score (expert13: 2.4 points on average) suggested that he might not have taken his aim seriously enough were also removed from the study; notably, these data did not adequately reflect the neural activity during archery. Finally, 26 subjects remained (13 elite archers and 13 expert archers), each of them provided us with an average of 160 s resting state EEG data as well as with EEG data during the aiming period corresponding to 30 trials (3 s for each trial; overall removal rate ≈ 20.8%).

### 2.4. Microstate Analysis

In the microstate analysis, the global field power (*GFP*) of all subjects in the two groups, at all time points during the resting and the aiming states, was calculated. The GFP is defined as follows:−
(1)$$GFP=\sqrt{{\sum_{i=1}^{C}\left(V_{i}\left(t\right)-V_{mean}\left(t\right)\right)}^{2}/C},$$
where *C* is the number of electrodes, $V_{i}\left(t\right)$ represents the potential value of the *i*-th electrode at time *t*, and $V_{mean}\left(t\right)$ represents the average potential of all electrodes at time *t*. The *GFP* was the standard deviation of potential of all channels at each time obtained by subtracting the average potential of all channels at each time from the potential of each channel at each time; the latter reflects the degree of the potential change between the given electrodes at a given time (Michel and Koenig 2018). Previous studies have shown that the potential distribution at the local maximum value of the *GFP* curve would maintain a stable state and express the highest signal-to-noise ratio (SNR) [56,57,58]. Thus, using the topographic map at the maximum point of the *GFP* to represent the topographic map around it is an effective method for the improvement of the microstate SNR and the reduction of the amount of calculation required. Through calculation, the *GFP* was obtained, and so were the corresponding “original maps” at the local *GFP* peaks.

We applied the atomize and agglomerate hierarchical clustering (AAHC) algorithm to cluster the “original maps”. The AAHC algorithm ignores the polarity of the potential topographic maps, and it is a “bottom-up” hierarchical clustering method with high efficiency. The algorithm considered each original potential topographic map as a class; it then calculated the spatial correlation between each potential topographic map and other topographic maps, identified the potential topographic map with the lowest global explained variance (*GEV*), and assigned it to the category with the highest correlation. Subsequently, the iteration was performed by removing one class at a time until a given number of potential topographic maps (i.e., the set number of microstate clusters) were obtained [59]. The calculation formula of GEV is as follows:(2)$${GEV}_{n}=\frac{{\left(GFP_{n}\cdot Corr\left(x_{n},a_{ln}\right)\right)}^{2}}{\sum {}_{n’}^{N}GFP_{n’}{}^{2}},$$
where ${GFP}_{n}$ is the global field power, which is calculated as the standard deviation across all electrodes of the EEG for the *n’*th time sample, *N* is the number of time samples, $x_{n}$ represents the *n’*th time sample of the recorded EEG and $a_{ln}$ signifies the topographical map assigned to nth EEG sample. $Corr\left(x_{n},a_{ln}\right)$ is the spatial correlation between data and the template map.

In this study, cross-validation (*CV*) criteria were used to determine the optimal number of microstate categories. The formula for CV is as follows [60]:(3)$$CV={\hat{\sigma }}^{2}\cdot {\left(\frac{C-1}{C-K-1}\right)}^{2},$$
where *K* is the number of clusters (microstate classes), $\hat{\sigma }$ is an estimator of the variance of the residual noise calculated as:(4)$${\hat{\sigma }}^{2}=\frac{\sum {}_{n}^{N}x_{n}{}^{T}x_{n}-{\left(a_{ln}{}^{T}\cdot x_{n}\right)}^{2}}{N\left(C-1\right)}.$$

A range of two to six microstate classes was set in the experiment, and the optimal number of classes in this experiment was determined to be 4 through *CV*. The scalp potential topography of each subject at each time point was compared with the microstates obtained by clustering, and the original maps were matched according to their correlation with the corresponding labels of the microstates (MS A–D).

In this study, four electroencephalogram microstate time series parameters (namely, duration, occurrence, coverage, and transition probability) were calculated to quantify the activity of each subject’s brain in the four microstate classes under the examined states. Duration is the average length of time for which each microstate remains stable, occurrence is the average number of microstates per second, coverage is the percentage of the specified microstate in the total recording time [61], and transition probability is the probability of changing from the current microstate to another microstate [62]. The aforementioned microstate analysis process was performed through the MATLAB Microstates 1.2 toolbox [63].

### 2.5. Sources of Microstates

The sLORETA is a method suitable for the estimation of the probable source of EEG signals in standard brain map spaces using a finite inversion algorithm. By calculating the inverse solution of the weighted minimum norm on the basis of the brain discrete distribution model, the EEG microstates are identified in the corresponding cortical electrical location with the maximum current density [53,64]. The electrode coordinates and the head models are based on the average MRI brain template devised by from Montreal Neurological Institute (MNI) [65]. As a real-time, linear, three-dimensionally distributed, and discrete EEG imaging method, sLORETA has a spatial resolution of 5 mm and can partition the brain volume into 6239 voxels, with a perfect first-order localization [53,66]. Meanwhile, sLORETA also has the advantages of being a low-cost and easily accessible method [51]. Therefore, the sLORETA map represents the standard electrical activity of each voxel in the MNI space and can be used to estimate the current density as well as to accurately obtain the location of a single microstate source.

The “BRL–sLORETA norms 2008” software (hereinafter referred to as sLORETA software) was jointly developed by the Brain Research Laboratory (BRL), Department of Psychiatry, New York University School of Medicine, and the KEY Institute for Brain-Mind Research, University Hospital of Psychiatry, Zurich [66]. In this experiment, the sLORETA software was used for the source reconstruction of all the microstates of the two groups of archers in both the resting and aiming states, and the results of the source analysis in each microstate were obtained, thus laying the foundation for further exploring of the physiological significance of the brain regions corresponding to the sources of these microstates.

### 2.6. Statistical Analysis

The Kolmogorov–Smirnov test (K–S test) was used to evaluate whether the EEG microstate parameters of the two groups of subjects in the examined states followed a normal distribution. It was found that not all parameters followed a normal distribution (*p* < 0.05), and so we uniformly adopted the Wilcoxon rank-sum test as one of the nonparametric tests allowing us to reliably analyze the data [67]. The false discovery rate (FDR) method was used to correct the results of the statistical tests. According to the experimental results, the *p* values of 0.05 and 0.001 were set as the limits of significant difference and very significant difference between the two groups of samples, respectively.

In the correlation test between the EEG microstate parameters and the archery performance, the K–S test was also applied on the average scores of each archer in the two groups of 30 shots. The test results showed that the average archery performance did not obey the normal distribution (*p* < 0.05), and for this reason, the Spearman rank correlation analysis was selected for the undertaking of the correlation analysis, and the correlation coefficient (*r*) was calculated. Similarly, we applied FDR to correct the *p* values. The statistical analysis part was conducted through the statistical test toolbox of the MATLAB R2014a platform.

## 3. Results

Four microstate maps of the two groups of archers during resting and aiming were identified through the AAHC analysis. The microstate analysis generally ignores potential polarity. According to the results in Figure 3, the microstate maps of the two groups in the resting state are relatively similar and consistent with the four archetypal microstate maps (namely, the right-frontal left-posterior, the left-frontal right-posterior, the midline frontal-occipital, and the midline frontal topographies). By contrast, the microstate maps during the aiming period are more irregular and significantly different from the archetypal microstate maps.

### 3.1. Microstate Duration

According to the statistical test results of the duration of each of the EEG microstates, as shown in Figure 4a, the microstates A, B, and D of the two groups during the resting state exhibited identified very significant differences (MS A: *p* = 3.5 × 10^−4^; MS B: *p* = 2.0 × 10^−8^; MS D: *p* = 9.4 × 10^−4^), and the microstates C between the two groups demonstrated significant differences (*p* = 9.4 × 10^−3^). Except for microstate D, all other microstates had a longer duration for the expert archers. Figure 4b presents the differences in the average duration of each microstate between the two groups of archers during the aiming period. Among them, the average duration of microstates A and B exhibited significant differences (MS A: *p* = 3.8 × 10^−6^; MS B: *p* = 0.002).

Table 2 presents the results of the change in the microstate duration during the aiming state and as compared with the resting state. It was found that the microstate A of the elite archers was significantly shorter than that of the resting state during the aiming period (*p* = 0.013) and that the microstate D is longer during the aiming period (*p* = 0.007). The duration of microstates B and C was found to be significantly decreased (MS B: *p* = 1.8 × 10^−11^ MS C: *p* = 1.4 × 10^−7^), whereas that of microstate D was significantly increased in the expert archers when compared with the resting state respective one (*p* = 1.1 × 10^−9^).

### 3.2. Occurrence of Microstates

Figure 4c compares the average occurrence of microstates between the two groups of archers during the resting state. The two groups of archers exhibited very significant differences only regarding microstate D (*p* = 5.7 × 10^−12^), whereas elite archers were characterized by a higher frequency of occurrence per second. The average occurrence of microstates in the two groups during the aiming period is presented in Figure 4d. The average occurrence of microstates C and D in the elite archers is significantly higher than that in the expert archers (MS D: *p* = 0.006), whereas microstate C revealed a very significant difference (*p* = 5.7 × 10^−4^) in that respect. Conversely, microstate A revealed the opposite significant results (*p* = 0.018).

The comparison of the occurrence presented in Table 2 shows a significant decrease in the aiming microstate C (*p* = 2.4 × 10^−6^) and a significant increase in the aiming microstate D (*p* = 0.038) when compared with the resting state. Expert archers demonstrated higher microstates A and D during aiming, which were identified as significant (MS A: *p* = 0.036) and very significant (MS D: *p* = 1.5 × 10^−10^), respectively, whereas their microstate C presented with a lower frequency of occurrence (*p* = 1.9 × 10^−6^).

### 3.3. Coverage of Microstates in Total Time

To directly reflect the coverage of each microstate in the total time of the two groups of archers under the two different conditions as well as their statistically significant differences, we presented the coverage of each microstate in total time and the results with significant differences in Figure 5. The resting state test results revealed that there were significant differences between the two groups regarding microstates B and D (MS A: *p* = 1.2 × 10^−6;^ MS A: *p* = 1.5 × 10^−11^). Elite archers had a higher coverage of microstate D and a lower coverage of microstate B than expert archers in terms of total time. There were very significant differences (MS A: *p* = 3.9 × 10^−9^; MS A: *p* = 9.0 × 10^−5^) identified in microstates A and B between the two groups when aiming. Elite archers demonstrated a lower coverage of microstate A than expert archers, whereas expert archers exhibited a lower coverage of microstate B.

According to the comparison between the resting and the aiming state provided in Table 2, a very significant difference (MS A: *p* = 0.008; MS C: *p* = 2.9 × 10^−6^ MS D: *p* = 3.8 × 10^−7^) was found regarding the microstates A and C of the elite archers. Consistently with the results of the above two parameters, microstate A and C was found to decrease, whereas microstate D was found to increase when aiming. Expert archers demonstrated significant differences regarding microstates B, C, and D (MS B: *p* = 3.8 × 10^−15^; MS A: *p* = 3.0 × 10^−8^ MS A: *p* = 8.8 × 10^−17^). The coverages of microstates B and C in terms of total time during aiming were lower than the respective ones in the resting state, whereas that of microstate D was higher.

### 3.4. Transition Probability between Different Microstates

Table 3 presents the significant difference of transition probability between different microstates during the resting and the aiming state, as well as between the corresponding microstates under different states between elite and expert archers. As shown in Table 3, the transfer probability of the EEG microstates between the two groups of archers during the resting state (with eyes closed) was very significantly different from that of microstate B to C (*p* = 2.3 × 10^−6^), microstate C to B (*p* = 7.9 × 10^−5^), microstate C to D (*p* = 5.9 × 10^−11^), and microstate D to C (*p* = 1.1 × 10^−9^). During the aiming state, there were significant differences in the transition probabilities between the two groups of archers between microstate B to A (*p* = 0.006), microstate B to C (*p* = 0.007), microstate C to D (*p* = 0.002), microstate D to A (*p* = 0.006), microstate D to B (*p* = 0.004), and microstate D to C (*p* = 0.005), and there were very significant differences in the transition probabilities between microstate A to B (*p* = 4.3 × 10^−6^).

Meanwhile, we calculated the average transfer probability of other microstate classes to a certain microstate under the two states to compare the potential activation trends of each microstate in professional archers. During the resting state, the average probability of the four microstates of elite and that of expert archers transferring from other microstates were (in descending order) D (8.31%), A (8.05%), C (7.78%), B (7.59%) and A (8.50%), B (8.16%), C (7.79%), D (6.96%), respectively. The average probabilities of the four microstates transferring from other microstates during archery aiming were for A, D, B, and C: 8.62%, 8.55%, 7.86%, and 7.22% for the elite archers and 9.11%, 8.61%, 7.95%, and 6.46% for the expert archers, respectively.

According to Table 3, the comparison between the microstate transition probabilities of the archers during the resting and the aiming states shows that there are significant differences between elite archers in microstate A to B (*p* = 0.019), microstate A to C (*p* = 0.014), microstate D to B (*p* = 0.001), and microstate D to C (*p* = 0.046) under the different states, whereas the differences between the microstate A to D (*p* = 1.3 × 10^−8^), the microstate C to A (*p* = 6.9 × 10^−4^), as well as the microstate D to A (*p* = 8.5 × 10^−5^) are very significant. Expert archers demonstrated significant transition probabilities between all microstates except for microstates D to B. Among them, microstates B to A (*p* = 0.021), microstates B to D (*p* = 0.017), and microstates D to C (*p* = 0.039) exhibited significant differences, and the transition probabilities from microstate A to B (*p* = 1.2 × 10^−4^), microstate A to C (*p* = 5.3 × 10^−5^), microstate A to D (*p* = 1.5 × 10^−6^), microstate B to C (*p* = 8.3 × 10^−7^), microstate C to A (*p* = 4.8 × 10^−9^), microstate C to B (*p* = 1.5 × 10^−6^), microstate C to D (*p* = 1.0 × 10^−4^), and microstate D to A (*p* = 3.1 × 10^−11^) revealed very significant differences.

### 3.5. Correlation between Microstate Parameters and Archery Performance

The statistical results shown in Table 4 were obtained by performing a Spearman rank correlation analysis between the microstate parameters of each archer in the two groups during aiming and their average archery performance. As shown, only the duration and coverage in microstate C of the elite archers were significantly correlated with their archery performance (duration: *p* = 1.1 × 10^−9^; coverage: *p* = 1.1 × 10^−9^), and they were negatively correlated. Moreover, as the correlation test results of the transfer probability between the various microstates and the archery performance did not reveal any significance, we did not list these data in the paper.

### 3.6. Source Localization of Microstates

sLORETA was used to analyze the four microstates of the two groups of archers under the herein examined different states, and consequently, the standardized current density images of the cerebral cortex distribution were obtained. The results of the source localization analysis in the resting state are presented in Figure 6a,b. The microstates A to D of the elite archers corresponded to areas 38 (superior temporal gyrus, temporal lobe), 11 (rectal gyrus, frontal lobe), 31 (posterior cingulate, limbic lobe), and 30 (posterior cingulate, limbic lobe) of the Brodmann’s partition system, respectively. The microstates A to D of the expert archers corresponded to Brodmann areas 31 (cingulate gyrus, limbic lobe), 18 (lingual gyrus, occipital lobe), 7 (precuneus, parietal lobe), and 30 (parahippocampal gyrus, limbic lobe), respectively.

Figure 6c,d present the results of the source localization analysis during aiming. The four microstate classes of elite archers corresponded to the Brodmann areas 37 (fusiform gyrus, temporal lobe), 37 (fusiform gyrus, temporal lobe), 11 (superior frontal gyrus, frontal lobe), and 37 (fusiform gyrus, temporal lobe), respectively. The four microstates of expert archers corresponded to Brodmann areas 37 (fusiform gyrus, temporal lobe), 11 (superior frontal gyrus, frontal lobe), 11 (superior frontal gyrus, frontal lobe), and 29 (posterior cingulate, limbic lobe), respectively.

## 4. Discussion

This study aimed to compare the EEG microstate parameters of the alpha rhythm in elite and in expert archers during resting and aiming, as well as to explore the influence of these two states on the dynamic brain network of professional archers with different competitive levels.

### 4.1. Microstate Analysis of Elite and Expert Archers in Resting State

The EEG microstates represent quasi-stable transient modes of coordinating electrical activity on the surface of the cerebral cortex and contain important microscopic information in EEG signals [17]. The microstates A, B, C, and D observed by previous scholars in resting state fMRI studies correspond to the four functions of the phonological processing, the visual network partial cognitive control, the partial default mode network (DMN), and the dorsal attention system, respectively [21,32,68]. According to the results of our sLORETA analysis, the microstate A of elite archers mainly occurs in the left temporal region as related to phonological recognition, while the microstate B of expert archers mainly occurs in the occipital region as related to vision [69]. The physiological significance of these brain regions is consistent with the corresponding functions of the microstates. The sources of microstate C in the two groups were distributed in the posterior cingulate and the precuneus of the parietal lobe, respectively. Studies on DMN have concluded that it is distributed in the medial prefrontal cortex (mPFC), the anterior cingulate, the posterior cingulate and the precuneus, the angular gyrus, and other brain regions. These areas also include the brain regions of the traceability analysis of this experiment, a fact that—to a certain extent—is supportive of the results of our source analysis.

In the statistical test results regarding the resting state with closed eyes, we identified some significant indicators between the two groups of archers in terms of the parameters of duration, occurrence, and coverage. Microstate D, particularly, is the only microstate class that demonstrated significance in all the above three microstate parameters, and these microstates of the elite archers are higher than those of expert archers. Moreover, all of them have revealed very significant statistical differences. According to the results of the transition probabilities between the different microstates, the other three classes of microstates of the elite archers have exhibited the highest average transition probability to microstate D, whereas the expert archers have expressed the highest average transition probability from all other three classes of microstates to that of microstate A. From this perspective, the neural state of the brains of elite archers is preferentially shifted to the dorsal attentional system. Numerous previous studies have reported the importance of being highly focused and maintaining steady attention for the delivery of excellent performance in archery and related fine motor tasks [42,70,71,72]. Interestingly, the results of this study also seem to indicate that the dorsal attention system associated with microstate D is the most important factor for the recorded difference in the resting state athletic ability between the elite and the expert archers.

Dorsal attention network has functional specialization nodes that promote specific processes of attention control [73], which is considered to be related to the voluntary control of attention [32,74,75,76], and the network supports orienting [77]. The dorsal attention system is a functional system activated by attentional tasks [32]. Some fMRI studies suggest that the blood oxygen level dependent (BOLD) signals in the dorsal attention system increase during the undertaking of such tasks [78,79]. Demarin et al. (2014) have proposed the theory of functional neuroplasticity, which suggests that through continuous learning and memory, the specific functional state of the brain would lead to the establishment of permanent changes in synaptic relations between neurons due to functional neuroplasticity [40]. We believe that during the long-term training and consolidation process, the dorsal attention system is frequently activated and continuously strengthened by the archery-related cognitive tasks in both groups of archers who have received professional archery training. When combined with our experimental results regarding the three parameters in microstate D of the elite archers that showed a higher value, this may indicate that longer periods of specific skills training have resulted in the elite archers’ dorsal system being more frequently activated, and constantly strengthened. Thus, functional neural state change is more obvious, which makes them have better attention control, especially in oriented attention.

It has been found that a higher EEG power in the alpha band during the resting state implies a stronger neural synchronization in the resting state “baseline” and a better performance on the cognitive motor tasks [80]. Hence, the resting EEG reflects a “baseline” state of the brain [39,81]. Some scholars have proposed that the microstate D is task-based positive in their studies involving microstate analysis in cognitive manipulation tasks [32,82]. The latter may indicate that after longer training of archery skills, the neural states related to the dorsal attention system in the brain of elite archers have a higher “baseline” value and a relatively more active state. Consequently, they can achieve the neurological state required to perform an archery task with less cognitive effort than the expert archers, thus suggesting that elite archers are more able to concentrate during training and competition. We also think that this is a sign of the elite archers being more neuro-efficient.

### 4.2. Microstate Analysis of Elite and Expert Archers during Aiming

Many significant differences were also identified regarding the microstate parameters of the two groups of archers during aiming. Few scholars have already studied the EEG microstates during motor tasks, and the research on the EEG microstates implicated in the movement processes related to archery has not been scientifically conducted. Particularly, the microstate maps during aiming in this experiment are more irregular and significantly different from the four archetypal microstate maps obtained during the resting state as proved by previous studies, so it is difficult to identify the physiological functions related to them. Based on the rigor of academic research, this study did not introduce the previous analysis conclusion regarding the microstates in the resting state into the analysis of the archery process. Thus, it might not be as rigorous to attempt to undertake further analysis from the point of view of significant differences between the various microstate parameters. We have, herein, mainly analyzed the results of archers during the aiming based on the physiological significance of the brain region where the microstate appears to originate from and based on the sLORETA source analysis.

The average transition probability from other microstate classes transferring to microstate A was the biggest for both groups of archers during aiming. Meanwhile, the source analysis results of microstate A in both groups of archers indicated the Brodmann area 37 of the left temporal region. This finding indicates that the neurological state of the professional archers is preferentially transferred to microstate A (i.e., the potential activation trend of the alpha rhythm in the left temporal region is stronger in both groups). In the source analysis during aiming, except for microstate A of both groups of archers, microstates B and D of the elite archers were also found to be distributed in the Brodmann area 37 of the left temporal region. The Brodmann area 37 has been associated with semantic language function [83,84]. From this point of view, the function corresponding to the microstate does not seem to be directly related to the aiming process. Our experiments analyzed the EEG microstates under the alpha rhythm, which mostly appeared in brain regions that did not participate in the task, thus reflecting the suppressed activity of this brain region [84,85]. Previous studies have demonstrated significant increases in alpha wave power in the left temporal region in professional athletes when undertaking cognitive motor tasks [47,48]. This also seems to support our experimental results. It may mean that both of them are professional archers, and their brains can reasonably allocate the functional state during aiming, so that the archer can focus on activating other brain regions associated with archery by inhibiting the function of word recognition and language comprehension in the left temporal region (which is not related to archery).

The results of our correlation test revealed a significant negative correlation between the duration and coverage of the elite archers’ microstate C and their archery performance. This suggests that the shorter the duration and the smaller the coverage (in terms of total time) of microstate C for elite archers are, the better their archery performance is. According to the values of the microstate parameters during aiming presented in Table 2, the three microstate parameters of microstate C for the elite archers are the lowest among all the microstate classes; a finding that seems to prove the statistical results of the aforementioned correlation. The source of elite archers’ microstate C, as calculated via sLORETA, occurs in the frontal lobe; the latter is believed to play an important role in the top-down regulation affecting visuospatial attention, working memory, as well as visual characteristic processing [86,87,88]. Considering that we were studying the alpha rhythms that are known to inhibit brain regions, we think this could also be explained by the elite archers effectively promoting attention by reducing the alpha rhythms’ inhibition of spatial attention, visual processing, and other functions in the frontal brain regions. This further proves that attention plays an important role in the successful completion of archery.

By observing the results of the source analysis performed through sLORETA in microstate D, we found that the source distribution of microstate D of the two groups of archers in the examined two states was in the limbic lobe except for the elite archers during aiming. Although a study has suggested that the limbic lobes of elite and expert archers were activated during tasks [55], we have yet to find a sufficient explanation for the specific connection between attentional function and the limbic lobe in other physiology-related studies. The latter may be due to the limited number of top archers being available as subjects.

Additionally, although we have listed in the results the changes of each corresponding microstate parameter between the two different states, when considering that the microstate maps during aiming are also quite different from those in the resting state. Likely, the corresponding microstates in the two states cannot be divided into the same classes, and they might represent different physiological significances. Thus, we did not attempt to clearly explain the parametric changes of each microstate category under the studied states but only took these results as a reference for future EEG microstate studies in similar cognitive motor tasks.

## 5. Conclusions

In this study, the resting state results demonstrate that the duration, frequency, and coverage of microstate D in elite archers are significantly higher than those in expert archers and that the transition probability of other microstate classes transferring to microstate D is the highest. During aiming, the average transition probability of other microstates transferring to microstate in the left temporal region was the highest observed in the two groups, and there was a significant negative correlation between the elite archers’ microstate in the frontal region and their archery performance. We hypothesize that attention control is the key differentiating factor between elite and expert archers. These conclusions confirm the hypothesis that the microstate parameters of the elite and the expert archers are significantly different under the different examined states and that the microstate parameters in the aiming state are closely related to archery performance itself. Our findings provide a valuable reference for the future study of EEG microstate dynamics in different brain states of professional athletes who perform cognitive motor tasks for a long time.

The study is a preliminary attempt to study the brain microstates of professional archers by physiological measurement. Although EEG microstate is a valuable and promising tool to quantify cognitive state and establish the relationship with behavior, it is necessary to say that the theoretical basis and standard of measurement for the EEG microstates during cognition tasks has not been effectively proved. Particularly, the physiological significance of the microstates during the task in motion-related fields remains to be further explored. In order to more strictly evaluate the psychological activities in cognitive tasks, in future research on the brain and neural mechanisms in fine movements related to archery behavior, a variety of physiological measurement techniques (such as eye tracker, fNIRS, heart rate monitor, etc.) should be combined with cognitive scale tests to comprehensively evaluate cognitive performance such as fatigue, arousal, working memory, etc. In addition, there should be a larger sample size and a variety of subjects with different competitive levels as the experimental objects. Furthermore, the design should include a more realistic competitive environment as the experimental conditions, so as to obtain more accurate research results to compare and analyze the neurophysiological activities related to shooting behavior under real conditions.

## Figures and Tables

**Figure 1 brainsci-12-01017-f001:**
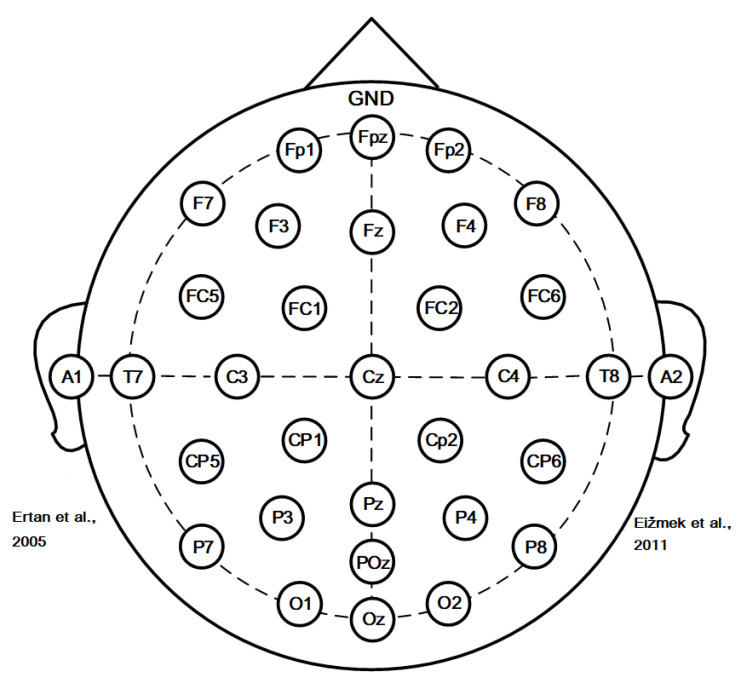
Placement of electrodes using the standard 10–20 system (32 channels: Fp1, Fpz, Fp2, F7, F3, Fz, F4, F8, FC5, FC1, FC2, FC6, T7, C3, Cz, C4, T8, CP5, CP1, CP2, CP6, P7, P3, Pz, P4, P8, POz, O1, Oz, and O2; forehead: ground; average left and right mastoid processes: [1,2]).

**Figure 2 brainsci-12-01017-f002:**
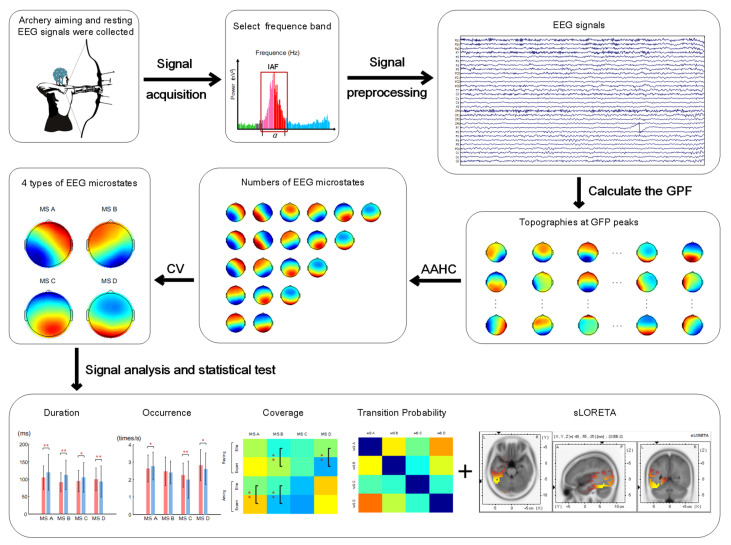
Synopsis of the undertaken experimental analysis process. Asterisks represent significant results in statistical test (*: *p* < 0.05; **: *p* < 0.001).

**Figure 3 brainsci-12-01017-f003:**
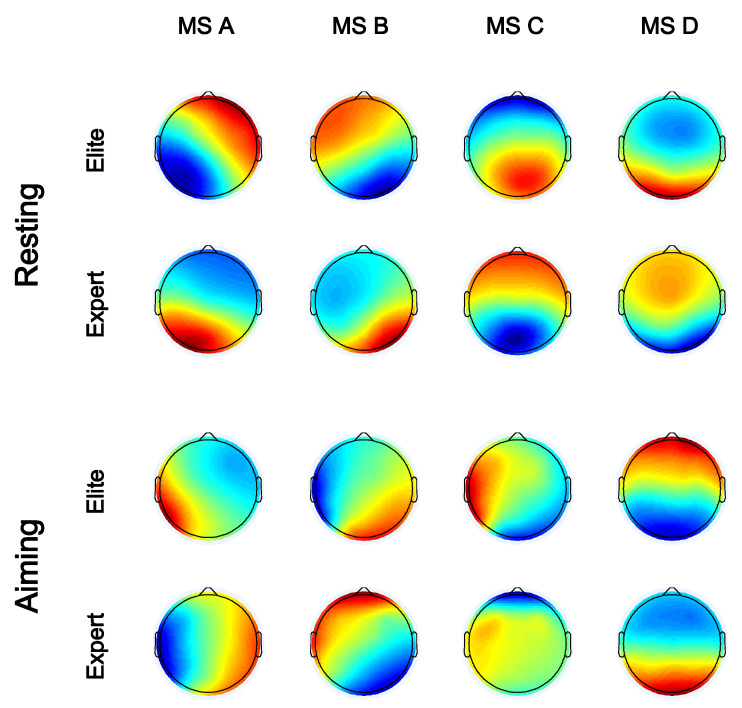
Four microstate maps of the elite and the expert archers, during the resting state with their eyes closed and during the aiming states.

**Figure 4 brainsci-12-01017-f004:**
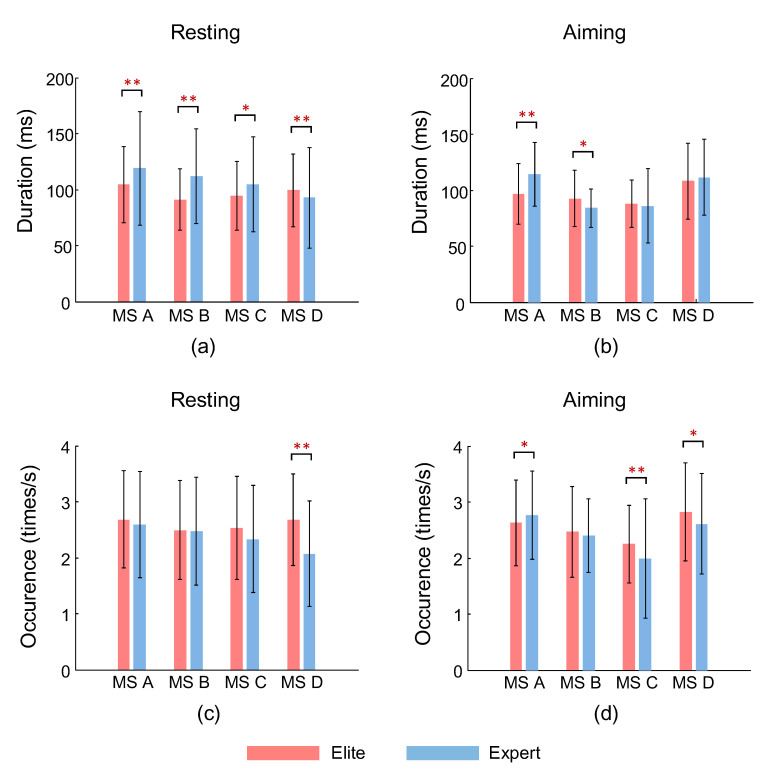
(**a**,**b**) present the average duration of the four microstates in the two groups during the resting with eyes closed and the aiming states. (**c**,**d**) present the mean and standard deviation of the average occurrence of the four microstate categories in the two groups under the conditions of eye-closing resting and aiming, respectively. The asterisk (s) at the top indicate (s) significant (*: *p* < 0.05) and very significant (**: *p* < 0.001) differences between the two groups of archers.

**Figure 5 brainsci-12-01017-f005:**
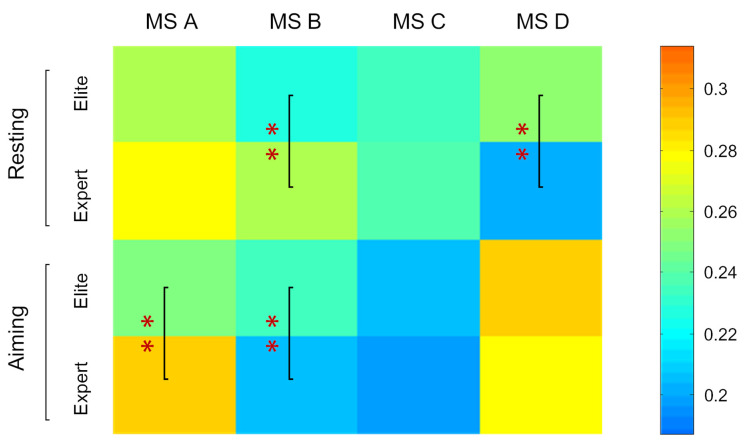
Average coverage of the four microstate classes in total time for the two groups of archers during the resting (with eyes closed) and the aiming states. The asterisks indicate a very significant difference between the two groups’ mean values (**: *p* < 0.001). The color indicator bar on the right represents the proportion of the average time coverage of each microstate in terms of total time. The orange color indicates the higher proportion of the microstate classes in terms of total time, and the blue color indicates the lower.

**Figure 6 brainsci-12-01017-f006:**
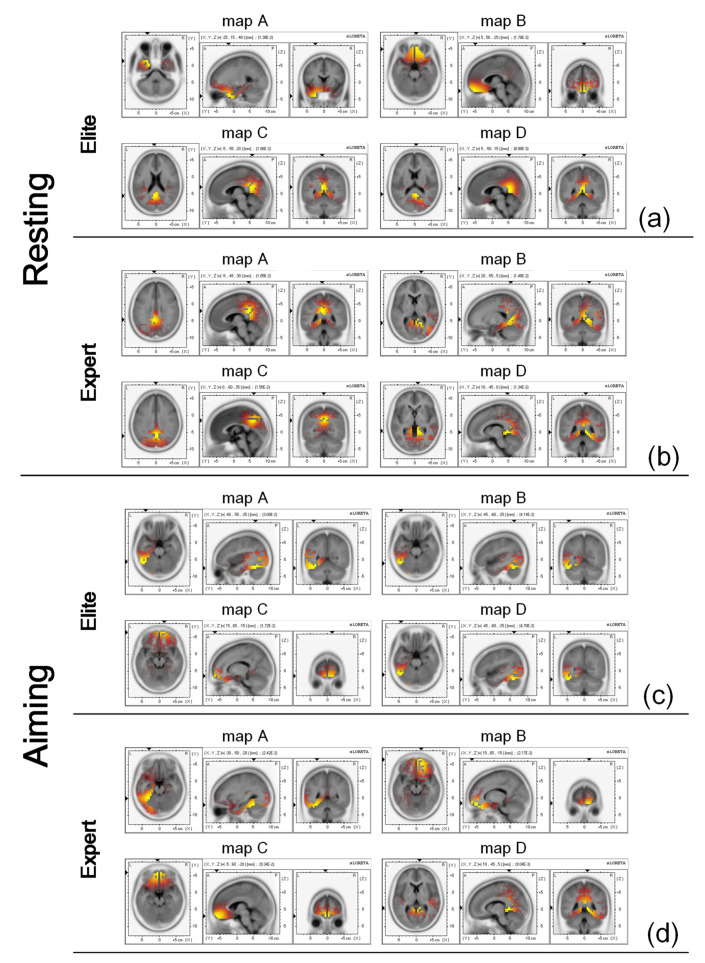
sLORETA was used for the source location analysis of the elite and expert archers’ data during the resting and aiming states. The colored areas represent the maximum current density, and the yellower the color, the higher the degree. (**a**) Topographic maps of the microstate source localization for the elite archers at the resting state. (**b**) Topographic maps of the microstate source localization for the expert archers at the resting state. (**c**) Topographic map of the microstate source localization for the elite archers during aiming. (**d**) Topographic map of the microstate source localization for the expert archers during aiming.

**Table 1 brainsci-12-01017-t001:** Demographic, training, and archery performance data of the recruited subjects.

Elite					Expert				
Subject	Age	Sex	Training Years	Archery Performance	Subject	Age	Sex	Training Years	Archery Performance
Elite1	28	Male	6.0	8.9	Expert1	25	Male	3.0	7.0
Elite2	24	Male	11.0	8.3	Expert2	22	Female	6.0	7.0
Elite3	22	Female	8.0	7.7	Expert3	24	Male	4.0	7.4
Elite4	22	Male	8.0	8.7	Expert4	21	Male	6.0	7.3
Elite5	20	Female	6.0	8.8	Expert5	26	Female	4.0	7.5
Elite6	23	Male	9.0	9.1	Expert6	23	Male	4.0	8.2
Elite7	22	Female	6.0	8.8	Expert7	26	Female	8.0	6.7
Elite8	26	Male	12.0	9.2	Expert8	24	Male	4.0	5.2
Elite9	26	Female	10.0	8.8	Expert9	23	Female	5.0	7.2
Elite10	23	Male	10.0	9.2	Expert10	22	Female	4.0	7.0
Elite11	23	Male	9.0	8.3	Expert11	22	Male	5.0	2.4
Elite12	24	Female	7.0	9.4	Expert12	20	Female	2.0	7.5
Elite13	22	Female	5.0	8.0	Expert13	19	Male	3.0	7.8
Elite14	18	Male	7.0	9.0	Expert14	25	Male	4.0	8.1
Elite15	23	Male	8.0	9.3	Expert15	23	Male	3.0	6.2
Elite16	22	Male	7.0	7.7					

**Table 2 brainsci-12-01017-t002:** Duration, occurrence, and coverage of each microstate in the resting and the aiming states of the elite and expert archers, as well as changes in the microstate parameters during aiming with respect to the resting state.

	Elite		*p*-Values Corrected by FDR	Expert		*p*-Values Corrected by FDR
	Resting Mean ± SD	Aiming Mean ± SD		Resting Mean ± SD	Aiming Mean ± SD	
**Duration (ms)**						
MS A	104.70 ± 34.01	95.77 ± 37.3	*↘	119.06 ± 50.59	114.09 ± 28.37	—
MS B	91.17 ±27.50	92.65 ± 24.99	—	112.15 ± 40.85	84.03 ± 17.06	**↘
MS C	94.76 ± 30.58	88.63 ± 20.895	—	104.66 ± 42.44	85.88 ± 33.11	**↘
MS D	99.43 ± 32.24	108.43 ± 33.98	**↗	92.87 ± 44.90	111.51 ± 33.97	**↗
Occurrence (times/s)						
MS A	2.69 ± 0.86	2.63 ± 0.76	—	2.59 ± 0.94	2.76 ± 0.78	*↗
MS B	2.50 ± 0.88	2.47 ± 0.81	—	2.48 ± 0.97	2.40 ± 0.66	—
MS C	2.53 ± 0.91	2.25 ± 0.70	**↘	2.33 ± 0.96	1.99 ± 1.06	**↘
MS D	2.68 ± 0.82	2.83 ± 0.88	*↗	2.07 ± 0.94	2.61 ± 0.90	*↗
Coverage (%)						
MS A	27.49 ± 11.78	25.32 ± 12.14	*↘	29.61 ± 13.53	31.41 ± 11.73	—
MS B	22.67 ±11.09	23.73 ± 12.92	—	27.14 ± 13.30	19.93 ± 7.63	**↘
MS C	23.83 ± 11.74	19.83 ± 8.33	**↘	24.01 ± 12.77	18.75 ± 14.83	**↘
MS D	26.01 ± 10 28	31.12 ± 15.18	**↗	19.24 ± 12.56	29.91 ± 14.03	**↗

MS represents microstate; *: *p* < 0.05; **: *p* < 0.001; ↗ and ↘ indicate the significant increase or decrease in each microstate parameter in the aiming state as compared with the resting state; — indicates that the microstate parameters show no significant difference in the aiming state in comparison with the resting state.

**Table 3 brainsci-12-01017-t003:** Mean and standard deviation of the transition probability between the microstates of the elite and expert archers during the resting and the aiming states (without considering the self-transition), and the *p*-value obtained by the Wilcoxon rank-sum test.

	Resting		Aiming		*p*-Values Corrected by FDR					
Microstate Transition	Elie (%) Mean ± SD	Expert (%) Mean ± SD	Elite (%) Mean ± SD	Expert (%) Mean ± SD	Elite vs. Expert		Resting vs. Aiming			
					Resting	Aiming	Elite		Expert	
A → B	8.66 ± 4.11	7.85 ± 4.77	7.63 ± 3.30	9.53 ± 3.51	0.089	∗∗	0.019 ∗	↘	**	↗
A → C	8.17 ± 3.73	9.56 ± 5.52	6.99 ± 2.56	7.26 ± 3.39	0.054	0.931	0.014 ∗	↘	∗∗	↘
A → D	7.64 ± 4.91	8.21 ± 4.27	10.51 ± 3.19	10.70 ± 5.68	0.553	0.985	∗∗	↗	∗∗	↗
B → A	7.89 ± 3.64	8.28 ± 4.57	8.21 ± 3.79	9.36 ± 3.34	0.779	0.006 ∗	0.919	—	0.021 ∗	↗
B → C	6.46 ± 3.46	8.29 ± 4.33	6.83 ± 3.05	5.86 ± 3.33	∗∗	0.007 ∗	0.333	—	∗∗	↘
B → D	8.48 ± 3.55	7.86 ± 4.75	8.38 ± 2.72	8.83 ± 3.44	0.235	0.512	0.962	—	0.017 ∗	↗
C → A	8.37 ± 3.75	9.66 ± 4.97	6.88 ± 2.61	6.59 ± 3.50	0.054	0.484	∗∗	↘	∗∗	↘
C → B	6.25 ± 2.85	8.68 ± 5.11	6.70 ± 3.09	6.10 ± 3.95	∗∗	0.053	0.333	—	∗∗	↘
C → D	8.45 ± 4.11	4.80 ± 3.30	7.82 ± 3.92	6.31 ± 2.77	∗∗	0.002 ∗	0.333	—	∗∗	↗
D → A	7.90 ± 3.94	7.57 ± 4.48	9.76 ± 3.49	11.38 ± 4.89	0.625	0.006 ∗	∗∗	↗	∗∗	↗
D → B	7.87 ± 3.71	7.95 ± 4.70	9.26 ± 3.02	8.22 ± 3.47	0.986	0.016 ∗	0.001 ∗	↗	0.380	—
D → C	8.72 ± 4.50	5.53 ± 3.58	7.59 ± 3.85	6.27 ± 2.86	∗∗	0.004 ∗	0.046 ∗	↘	0.039 ∗	↗

∗: *p* < 0.05; ∗∗: *p* < 0.001; ↗ and ↘ indicate the significant increase or decrease in each microstate parameter during the aiming state as compared with the resting state; — indicates that the microstate parameters show no significant difference in the aiming state when compared with those of the resting stat.

**Table 4 brainsci-12-01017-t004:** Correlation test results between microstate parameters and archery performance of elite and expert archers during aiming.

	Microstate Parameter	Correlation Indicator	MS A	MS B	MS C	MS D
Elite	Duration	*r*	0.299	−0.124	−0.798 ∗	0.250
		*p*	0.319	0.687	0.001	0.409
	Occurrence	*r*	0.294	0.217	−0.542	0.327
		*p*	0.329	0.476	0.056	0.275
	Coverage	*r*	0.314	0.066	−0.726 ∗	0.261
		*p*	0.297	0.830	0.005	0.388
Expert	Duration	*r*	0.135	−0.058	−0.182	0.028
		*p*	0.661	0.851	0.553	0.929
	Occurrence	*r*	0.300	0.072	0.085	0.030
		*p*	0.320	0.816	0.782	0.922
	Coverage	*r*	0.283	−0.038	−0.094	0.113
		*p*	0.348	0.908	0.761	0.714

MS represents microstate; ∗: *p* < 0.05; *r* is the correlation coefficient; *p* is the *p*-value corrected by FDR.

## Data Availability

The data that support the findings of this study are available from the corresponding author upon reasonable request.

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
