# Peer review of "Research on Top Archer’s EEG Microstates and Source Analysis in Different States"

_brainsci, 2022, doi:10.3390/brainsci12081017_

Round 1

Reviewer 1 Report

Authors in this paper indicate that elite archers are more active in the dorsal attention system and demonstrate a higher neural efficiency during the resting state. An interesting conclusion is that when aiming, professional archers experience an activation of brain regions associated with archery by suppressing irrelevant to the task brain regions. Undoubtedly, authors findings provide a novel theoretical basis for the study of EEG microstate dynamics in archery and related cognitive motor tasks, particularly from the perspective of the subject’s mental state.

My comments to the article are as follows:

- I propose a slight modification of the title of the article, because now it sounds a bit more like the title of a popular science article, not a strictly scientific article. I propose to remove the question: "From expert to elite?"

- I propose to expand the background of the article by referring to the implementation of the sLORETA method to date in EEG research. For example, by referring to: Using the LORETA method for localization of the EEG signal sources in BCI technology, Analysis and Classification of EEG Signals for Brain – Computer Interfaces, Series Title: Studies in Computational Intelligence, Springer 2020

- In my opinion, the descriptions of axes in the drawings should be corrected. They are currently chaotic. For example in Fig. 4. I think the wrong font was used when generating the PDF.

- I propose to extend point "2.4. Microstate Analysis" for more details on the mathematical description. Currently, there is only one formula.

- I propose to change the title of the section: "Conclusion" to "Conclusions"

- As part of Conclusions, I propose to write about plans for the future in the field of this research topic.

Author Response

We sincerely show the gratitude to the Editor and Reviewers for their careful and constructive comments on the manuscript. We have addressed all the comments and made changes as appropriate. The revised parts of the paper are marked in yellow.

Q1I propose a slight modification of the title of the article, because now it sounds a bit more like the title of a popular science article, not a strictly scientific article. I propose to remove the question: "From expert to elite?"

Answer: Thanks for your advice. We have revised the title of the paper to remove the " From expert to elite?".

Q2I propose to expand the background of the article by referring to the implementation of the sLORETA method to date in EEG research. For example, by referring to: Using the LORETA method for localization of the EEG signal sources in BCI technology, Analysis and Classification of EEG Signals for Brain – Computer Interfaces, Series Title: Studies in Computational Intelligence, Springer 2020.

Answer: Thank you for pointing out the shortcomings of our paper. We totally agree with you on our thesis. We further expand the relevant background knowledge of sLORETA in the Introduction section. The references mentioned by the reviewers and other research literatures on the application of sLORETA to EEG and BCI were added. This part of the content is in lines 169 to 173 on page 4 of the paper.

Q3: In my opinion, the descriptions of axes in the drawings should be corrected. They are currently chaotic. For example in Fig. 4. I think the wrong font was used when generating the PDF.

Answer: Thanks for your careful and comprehensive review of this manuscript. We carefully checked the pictures in our paper. However, after inspection, no confusion was found in the drawings we submitted. For the problem you mentioned in Fig. 4, we will explain the coordinate axis description here. The horizontal axis represents different microstate categories (i.e. microstate a to D, “Ms” represents microstate, which is mentioned in Table 2). The "times/s" of the vertical axis represents the occurrence frequency of the micro state, that is, the number of times it occurs per second. If you are still dissatisfied with our pictures, we will try our best to modify them until you agree.

Q4: I propose to extend point "2.4. Microstate Analysis" for more details on the mathematical description. Currently, there is only one formula.

Answer: Thanks for your careful and comprehensive review of this manuscript. We agree with you. In order to facilitate readers to understand our paper. We have added the formulas of GEV and CV and explained them to enrich the content of the paper. This part of the content is in lines 315 to 328 on page 8 and 9 of the paper.

Q5: I propose to change the title of the section: "Conclusion" to "Conclusions"

Answer: Thanks for your advice. We change "Conclusion" to "Conclusions".

Q6: As part of Conclusions, I propose to write about plans for the future in the field of this research topic.

Answer: Thank you for your advice. According to your suggestion, we put forward our views on the future research direction of EEG micro state in the field of archery related motion in the conclusion part. I hope this can provide some help for the follow-up research. This part of the content is in lines 703 to 718 on page 21 of the paper.

Through the revision of the paper, we have further understandings of the relevant knowledge, and also understand that there are many shortcomings in my research in the relevant field and need to be improved. Your final opinion is very important to us. If we still need to modify the manuscript, we will respect your opinion and resolutely correct it. Finally, once again sincerely thank you for your suggestions on our paper.

Reviewer 2 Report

The purpose of the present study was to compare some psychophysiological parameters (i.e. EEG microstates) of archers of different professional level, considering both the resting state and the actual motor act. For these reasons, the study was conducted with a portable device. The results revealed the presence of some significant differences between the two groups both regarding the resting state, showing a different engagement at the level of the dorsal attention system (microstate D), and during aiming, finding a different recruitment of the left temporal region.

The study is interesting and quite novel, however I have few concerns that the authors should address for a clearer understanding.

- grammar and english language revision is required

- the abstract should really be improved and made clearer also for the readers and researchers who are not familiar with the terminology (who are elite archers, for example? it should be evident from the very beginning)

- please be careful when talking about biofeedback and neurofeedback, please check on page 2 because it is a little bit confusive

- the use of LORETA with these few electrodes is considered a bit controversial. In fact, it is usually recommended for high-density caps. For this reason, the authors should provide a reference proving its feasibility and utility. Also, the claim they make in the discussion and conclusions should be way more cautious because LORETA with 32 electrodes is quite an aproximation, so I would suggest to adopt a more cautious approach.

- my biggest concern is that the paper is really technical and does not provide a strong theoretical framework that other researchers using different techniques could use to guide their work on archery or other sports. For example: what does it mean that the elite group have a higher engagement of the dorsal attentional system? The claim that time could have enhanced the neural circuits should be at least deepened and anticipated in the introduction, as a guiding hypothesis. Also, what do the authors mean with "... the professional archers can reasonably allocate the functional state of their brain during aiming by inhibiting the function of word recognition and language comprehension in the left temproal ragion (which is not related to archery), to better focus on activating the brain regions associated with archery"? This would be valid for all the other brain areas, so I believe this is a weak hypothesis that should be reformulated. 

All the results should be properly explained by making reference to shared cognitive theories that even people using other techniques (eye-tracker, fNIRS, behavioral measures) could use to guide their experimental apparatus, as well as to interpret results. Otherwise, the new knowledge that this paper brings remains mainly anchored to the area of microstates, the application of which, by the authors' own admission, has not yet been extensively tested and validated. For this reason, the technique would benefit from more theoretical reference support. 

Author Response

We sincerely show the gratitude to the Editor and Reviewers for their careful and constructive comments on the manuscript. We have addressed all the comments and made changes as appropriate. The revised parts of the paper are marked in yellow.

Q1grammar and english language revision is required.

Answer: Thanks for your advice. Since English is not our mother tongue, after the paper was revised, we sought professional help to polish our grammar, spelling and tense. The language of the article is more fluent and easy to understand. We submitted the editing certificate of the paper at the same time of submitting the revised paper.

Q2the abstract should really be improved and made clearer also for the readers and researchers who are not familiar with the terminology (who are elite archers, for example? it should be evident from the very beginning).

Answer: Thanks for your careful and comprehensive review of this manuscript. We have revised the abstract of the paper to further clarify the personnel information of the two groups of shooters, so that readers can understand the specific groups referred to by elite and expert shooters in the paper when reading the abstract. This part of the content is in lines 20 to 22 on page 1 of the paper.

Q3: please be careful when talking about biofeedback and neurofeedback, please check on page 2 because it is a little bit confusive

Answer: Thanks for your careful and comprehensive review of this manuscript. We reread the literature and revised this part to make the expression of neural feedback more rigorous. This part of the content is in lines 55 to 68 on page 2 of the paper.

Q4: the use of LORETA with these few electrodes is considered a bit controversial. In fact, it is usually recommended for high-density caps. For this reason, the authors should provide a reference proving its feasibility and utility. Also, the claim they make in the discussion and conclusions should be way more cautious because LORETA with 32 electrodes is quite an aproximation, so I would suggest to adopt a more cautious approach.

A: Thanks for your advice. sLORETA is a classical method to solve inverse problems. Previous studies have usually applied sLORETA to EEG amplifiers with high-density electrodes. In order to make our research more rigorous, we also referred to some literatures using the combination of sLORETA and EEG technology in the process of writing this paper, and found that sLORETA can also be used in EEG amplifiers with relatively low density. For example, pizzagalli and others used LORETA technology to combine with ERP as early as 2000, which is reflected in the paper "face activated ERPs and effective attribute: brain electrical microscope and tomography analyses". The number of channels of EEG amplifier used in this paper is 27. Liu et al. Also combined sLORETA's method with microstate dynamics in patients with absence epilepsy (2021) in the paper "alternative peri seize EEG microstate dynamics", and the number of channels of EEG amplifiers used in this study is 19. In other EEG studies, Imperatori et al. and Dvořáčková et al. used the sLORETA technology in their papers "aberrant EEG functional connectivity and EEG power spectrum in resting state post traumatic stress disorder: a sLORETA study (2014)" and "options for studying human motion: neurological program sLORETA (2019)" respectively, and also applied it to EEG amplifiers with 19 channels. Cevada et al. Applied sLORETA technology to EEG amplifiers with 20 channels in their paper "resilience, psychological characteristics, and restoring state brain cortical activity in atmospheres and non athelets (2021)". These studies also seem to show that sLORETA is also applicable in EEG amplifiers with relatively low density. SLORETA is a mature method. It has been proved in some literatures that the combination of this method and EEG microstate analysis can effectively obtain the spatial information of these microstate categories (Pizzagalli, D., Lehmann, D., König, T., Regard, M., & Pascual-Marqui, R. D. (2000). Face-elicited ERPs and affective attitude: brain electric microstate and tomography analyses. Clinical neurophysiology111(3), 521-531./ Lehmann, D., Faber, P. L., Gianotti, L. R., Kochi, K., & Pascual-Marqui, R. D. (2006). Coherence and phase locking in the scalp EEG and between LORETA model sources, and microstates as putative mechanisms of brain temporo-spatial functional organization. Journal of Physiology-Paris99(1), 29-36./Gianotti, L., Faber, P., Pascual-Marqui, R., Kochi, K., & Lehmann, D. (2007). Processing of positive versus negative emotional words is incorporated in anterior versus posterior brain areas: an ERP microstate LORETA study. Chaos and Complexity Letter2, 189-211./ Liu, H., Tang, H., Wei, W., Wang, G., Du, Y., & Ruan, J. (2021). Altered peri-seizure EEG microstate dynamics in patients with absence epilepsy. Seizure88, 15-21.), which is also an important reason why we use sLORETA in this study.

In order to let readers see these research results related to sLORETA, we further expand the background knowledge of sLORETA in the introduction and add references to these documents. This part of the content is in lines 169 to 173 on page 4 of the paper.

Q5: my biggest concern is that the paper is really technical and does not provide a strong theoretical framework that other researchers using different techniques could use to guide their work on archery or other sports. For example: what does it mean that the elite group have a higher engagement of the dorsal attentional system? The claim that time could have enhanced the neural circuits should be at least deepened and anticipated in the introduction, as a guiding hypothesis. Also, what do the authors mean with "... the professional archers can reasonably allocate the functional state of their brain during aiming by inhibiting the function of word recognition and language comprehension in the left temproal ragion (which is not related to archery), to better focus on activating the brain regions associated with archery"? This would be valid for all the other brain areas, so I believe this is a weak hypothesis that should be reformulated.

A: Thank you very much for your targeted suggestions. As you mentioned, our paper is indeed weak in theoretical support. In view of these problems, we have modified the corresponding position of the paper. In the analysis part of the dorsal attention network theory, we have deepened our understanding of the dorsal attention network by further reading the literature, in which it is mentioned that the dorsal attention system regulates the voluntary allocation of attention to positions or features guided from top to bottom (Vossel, S., Geng, J. J., & Fink, G. R. (2014). Dorsal and ventral attention systems: distinct neural circuits but collaborative roles. The Neuroscientist20(2), 150-159.). This system is considered to be related to voluntary control of attention (Klingberg, T.; O'Sullivan, B.T.; Roland, P.E. Bilateral activation of fronto-parietal networks by incrementing demand in a working memory task. Cerebral cortex (New York, NY: 1991), 1997, 7 (5), pp. 465-471./ Mantini, D.; Perrucci, M.G.; Del Gratta, C.; Romani, G. L.; Corbetta, M. Electrophysiological signatures of resting state networks in the human brain. Proceedings of the National Academy of Sciences, 2007, 104 (32), pp. 13170-13175./ Ozaki, T.J. Frontal-to-parietal top-down causal streams along the dorsal attention network exclusively mediate voluntary orienting of attention. PLoS One, 2011, 6 (5), e20079./ Seitzman, B A.; Abell, M.; Bartley, S.C.; Erickson, M.A.; Bolbecker, A.R.; Hetrick, W.P. Cognitive manipulation of brain electric microstates. Neuroimage, 2017, 146, pp. 533-543.), and the network supports orienting (Marek, S.; Dosenbach, N.U.F. The frontoparietal network: function, electrophysiology, and importance of individual precision mapping. Dialogues in Clinical Neuroscience, 2018, 20 (2).). We summarized the above theories and introduced more relevant knowledge and references to explain the phenomenon that elite shooters have stronger dorsal attention networks. This part of the content is in lines 593 to 595 on page 18 and in lines 605 to 610 on page 19 of the paper.

As for the " The claim that time could have enhanced the neural circuits should be at least deepened and anticipated in the introduction, as a guiding hypothesis " you proposed, we believe that in terms of the brain neural mechanism of athletes, long-term training can cause changes in neuroplasticity, which enables athletes with stronger skills to have higher neural efficiency. We put that in the introduction and we further explain the two concepts. This part of the content is in lines 135 to 140 on page 3 of the paper.

In "... the professional archers can reasonably allocate the functional state of their brain during aiming by inhibiting the function of word recognition and language comprehension in the left temproal ragion (which is not related to archery), to better focus on activating the brain regions associated with archery", we hope to enrich the left temporal region with the help of previous research conclusions. The results of sLORETA in the paper are mostly distributed in the left temporal region. Because previous studies have also confirmed the phenomenon of alpha power increase in the shooter's left temporal region during shooting aiming, and believe that this is because alpha waves inhibit the physiological functions of the left temporal region that are not related to shooting behavior (the left temporal region is related to semantic understanding), so that the athlete's brain can focus more on the activation of brain regions related to shooting behavior. In fact, this phenomenon that the brain reasonably distributes energy to activate and inhibit brain regions with different functions is also a classic in the field of motor neuroscience. We think that this seems to explain the results of our experiment to a certain extent, and we want to support the view of our paper with the previous research conclusions. Of course, this is the commonness of the two. We just want to explain this phenomenon here, and do not attribute this part to the differences between experts and elite shooters. Therefore, we reorganize the language of this part so that readers can understand the analysis content of this part. This part of the content is in lines 652 to 656 on page 20 of the paper.

Once again, thank you for your careful review of our manuscript.

Q6: All the results should be properly explained by making reference to shared cognitive theories that even people using other techniques (eye-tracker, fNIRS, behavioral measures) could use to guide their experimental apparatus, as well as to interpret results. Otherwise, the new knowledge that this paper brings remains mainly anchored to the area of microstates, the application of which, by the authors' own admission, has not yet been extensively tested and validated. For this reason, the technique would benefit from more theoretical reference support.

A (6): Thank you for pointing out the shortcomings of our paper. We agree with you. The original intention of this paper is to explore the neural mechanism of the sports field (archery behavior) from the perspective of micro state analysis. In our view, the important innovation of this paper is the innovation of analysis methods, that is, the combination of micro state and competitive sports (static sports), which provides a new perspective for subsequent related research. Another published paper, "from expert to elite? - Research on top archer's EEG network topology", analyzes the brain network characteristics of top shooters from the perspective of brain network topology, and has also been accepted by SCI journals. And because our research object is the top shooter who has been trained for a long time, we hope to explore it from the perspective of neural plasticity. Neuroplasticity is usually manifested in the resting state. The neural mechanism of resting state is also the ideal state of micro state analysis. At the same time, previous studies on micro states in cognitive tasks mostly focused on the analysis of micro states during cognitive control and motor imagination. Therefore, we analyze it from two aspects: the resting state and the micro state during archery aiming, so as to obtain more sufficient experimental results. In order to ensure the preciseness of the paper content and analysis results, we try to choose a more reasonable and universal theory to support our experimental results. For this reason, our analysis results are mostly based on the previous research results of micro state in cognitive tasks, as well as the more general and classic research results of motor neuroscience, and we dare not presume. At the same time, there are few previous studies on the combination of microstate analysis and other biometric technologies, so it is difficult for us to explain or promote it in combination with other biometric technologies, because it seems to be imprecise. This may also be the main reason why our results benefit from more theoretical references.

We attach great importance to your comments and regard them as the direction of our next efforts. Unfortunately, we are researchers majoring in communication, so we pay more attention to the research on the methods of signal analysis and feature extraction in this paper, which may not be enough for the research on cognitive theory and biological anatomy, which is also the main aspect that we need to improve in the next step. Based on this, in the conclusion part, we further put forward the research plan of micro state in the field related to archery in the future, so as to better guide us and other interested scholars in the future research direction. I hope our views can win your support. Your recognition is very important to us.

During the revision, we may not have a thorough understanding of your intention. If there are still deficiencies in our paper after the revision, please point out our shortcomings, and we will try our best to revise your comments until we get your approval.

Through the revision of the paper, we have further understandings of the relevant knowledge, and also understand that there are many shortcomings in my research in the relevant field and need to be improved. Your final opinion is very important to us. If we still need to modify the manuscript, we will respect your opinion and resolutely correct it. Finally, once again sincerely thank you for your suggestions on our paper.

Round 2

Reviewer 1 Report

Dear Authors, 

Thank you for the changes made.

I accept the introduced amendments.

I recommend this paper for publication.

Reviewer 2 Report

 the manuscript have been improved including the answers to all my concerns.
The paper is ready for publication, however I just have to highlight again that the term "EEG biofeedback" should be modified in EEG-based neurofeedback. I already suggested this but probably I wasn't clear in which part of the page.

MEV